# A Novel Quantitative Computer-Assisted Score Can Improve Repeatability in the Estimate of Vascular Calcifications at the Abdominal Aorta

**DOI:** 10.3390/nu14204276

**Published:** 2022-10-13

**Authors:** Maria Fusaro, Enrico Schileo, Gianluigi Crimi, Andrea Aghi, Alberto Bazzocchi, Giovanni Barbanti Brodano, Marco Girolami, Stefania Sella, Cristina Politi, Serge Ferrari, Chiara Gasperini, Giovanni Tripepi, Fulvia Taddei

**Affiliations:** 1National Research Council (CNR)—Institute of Clinical Physiology (IFC), Via G. Moruzzi 1, 56124 Pisa, Italy; 2Department of Medicine, University of Padua, Via Giustiniani 2, 35128 Padova, Italy; 3Bioengineering and Computing Laboratory, IRCCS Istituto Ortopedico Rizzoli, 40136 Bologna, Italy; 4Department of Medicine, Clinica Medica 1, University of Padova, 35128 Padova, Italy; 5Radiology Unit, IRCCS Istituto Ortopedico Rizzoli, 40136 Bologna, Italy; 6Spine Surgery Unit, IRCCS Istituto Ortopedico Rizzoli, 40136 Bologna, Italy; 7CNR-IFC, Clinical Epidemiology and Pathophysiology of Renal Diseases and Hypertension, 89124 Reggio Calabria, Italy; 8Service des Maladies Osseuses, Département de Médecine, HUG, 1205 Genève, Switzerland

**Keywords:** abdominal aorta calcifications, bone fractures, lateral spine radiograph, minimum detectable difference, measurement precision, vascular calcification score

## Abstract

In CKD and in the elderly, Vascular Calcifications (VC) are associated to cardiovascular events and bone fractures. VC scores at the abdominal aorta (AA) from lateral spine radiographs are widely applied (the 0–24 semiquantitative discrete visual score (SV) being the most used). We hypothesised that a novel continuum score based on quantitative computer-assisted tracking of calcifications (QC score) can improve the precision of the SV score. This study tested the repeatability and reproducibility of QC score and SV score. In forty-four patients with VC from an earlier study, five experts from four specialties evaluated the data twice using a dedicated software. Test–retest was performed on eight subjects. QC results were reported in a 0–24 scale to readily compare with SV. The QC score showed higher intra-operator repeatability: the 95% CI of Bland–Altman differences was almost halved in QC; intra-operator R^2^ improved from 0.67 for SV to 0.79 for QC. Inter-observer repeatability was higher for QC score in the first (Intraclass Correlation Coefficient 0.78 vs. 0.64), but not in the second evaluation (0.84 vs. 0.82), indicating a possible heavier learning artefact for SV. The Minimum Detectable Difference (MDD) was smaller for QC (2.98 vs. 4 for SV, in the 0–24 range). Both scores were insensitive to test–retest procedure. Notably, QC and SV scores were discordant: SV showed generally higher values, and an increasing trend of differences with VC severity. In summary, the new QC score improved the precision of lateral spine radiograph scores in estimating VC. We reported for the first time an estimate of MDD in VC assessment that was 25% lower for the new QC score with respect to the usual SV score. An ongoing study will determine whether this lower MDD may reduce follow-up times to check for VC progression.

## 1. Introduction

Mineral deposits on arterial walls, known as vascular calcifications (VC), are highly prevalent in aging, chronic kidney disease (CKD) and metabolic syndrome [1]. Phosphatemia/Hyperphosphatemia and Vitamin K deficiency, two conditions closely related to nutrition, have been shown to be associated to VC and bone fractures in CKD [2]. VC can develop at the intima or media layers of arterial walls. Calcifications of the intima develop as plaques and occlusive lesions in common atherosclerosis. Calcifications of the media, known as Mönckeberg’s arteriosclerosis and mostly observed in arteries with prevalent muscle content in aging, diabetes, and CKD, are usually thin and concentric, not protruding into the arterial lumen [3]. Their radiological appearance is also different: discrete plaques with irregular distribution for intima lesions, linear deposits for media. VC erode aortic compliance and energy storage capabilities (i.e., the ability of aortic walls to elastically deform to accommodate e.g., pressure differences) [4]. In CKD patients and in the elderly, there is solid evidence that VC have prognostic value for future cardiovascular events and poorer prognosis [5]. The relationship between arterial calcification and bone physiology (known as the bone–vascular axis) [6] is supported by the association of VC with bone fractures [7,8,9,10], and by that of abdominal aorta calcifications (AAC) with vertebral fractures in particular [11,12].

There is therefore considerable interest in tracking the emergence/progression of VC. X-rays techniques are commonly used, due to the radiopacity of mineral deposits. The measurement of coronary arteries calcium (CAC) content from coronary multi-slice computed tomography (CT) images is the current gold standard [13,14]. CAC measurement is quantitative and clinically relevant, but unsuitable for screening or frequent follow-up assessments because of the high radiation dose and costs. An alternative is to derive VC scores from the calcifications visible in X-ray projections. Although generally less accurate, this has become a popular approach, because simple radiographs deliver a lower radiation dose than CT and are largely available. When looking at VC in medium-sized arteries it is more common to derive simple binary scores, which check just the presence vs. absence of calcifications [15]. Conversely, measurements of VC length can be attempted when looking at the abdominal aorta (AA) using a lateral X-ray projection (or even a dual-energy X-ray absorptiometry (DXA) scan) at the lumbar spine. Different AAC scores have been proposed in the nineties. Witteman et al. proposed to measure the overall calcified length in the aorta tract corresponding to the L1–L4 spinal segment, then score the calcification severity based on four ranges (<1 cm, 2–5 cm, 6–10 cm, >10 cm) [16]. In that study, however, the measurement method was not specified. Kauppila et al. proposed instead a semi-quantitative method, based on the visual assessment of calcification length. To assist the operator, the formulation of the Kauppila score divides the AA in four tracts delimited by L1–L4 intervertebral spaces, and the operator is asked to score each anterior and posterior intervertebral tract in tertiles (0–3) based on the amount of calcification visually estimated (absent = 0, less than 1/3 = 1, from 1/3 to 2/3 = 2, more than 2/3 = 3). The final score is obtained by summing all tertiles contributions on the eight AA tracts (anterior and posterior for four intervertebral spaces) yielding a final score that ranges from 0 (no calcifications) to 24 (fully calcified AA) [17]. We will refer to this score as to the semiquantitative, visually-assessed score (hereinafter SV score).

The SV score is currently the most used to assess AAC from radiographs. It has shown good intra-operator and satisfactory inter-operator intraclass correlation coefficient (ICC) (several studies summarized in Schousboe et al. [18]). In recent years it has been increasingly used in longitudinal studies to detect progression of AAC. It is important to notice that, in absence of specific guidelines, follow-up times shortened from the 25 years of the initial studies [17,19], to 3 years [20,21] or even 1 year [22,23]. Consequently, significant differences down to even only 1 score point (in the 0–24 scale) have been reported [23]. This shortening of follow-up times went beyond the only indication about minimum follow-up times in the literature (five years, in the review of Szulc in 2016 [9]). However, that statement was not supported by specific reference to any study addressing precision, accuracy, and clinical relevance of AAC follow-up.

In particular, no study to our knowledge has estimated the minimum detectable difference (MDD) between measurements, which is key to assess significance of changes [24]. Semiquantitative nature and visual estimation of the SV score are two factors that may contribute to worsen the MDD, especially when assessed by non-expert clinicians from different specialties. Images with sub-optimal quality and resolution, such as DXA, an increasingly adopted source for AAC scoring, may also worsen the MDD [18].

We here propose a novel, Quantitative and Computer-assisted AAC score (hereinafter QC score) and hypothesize it can improve the precision of the semiquantitative visual (SV) score. The aim of the present work is to comprehensively evaluate repeatability and reproducibility metrics, including MDD estimate, of this novel QC score and of the SV score. A computer-assisted version of the SV score (hereinafter SC score, for semiquantitative computer-assisted) will also be tested to eliminate the visual bias from the comparison.

## 2. Methods

### 2.1. Concept

The new score we propose (QC score) measures the relative calcified length of the abdominal aorta. QC is determined on digital radiographs of the lumbar spine through the computer assisted tracking of calcified tracts in the posterior and anterior arterial walls between the T12-L1 and the L4–L5 intervertebral spaces (see Section 2.4 for details). 

In a cohort of subjects with AAC, we assessed intra- and inter-operator repeatability, and MDD of: (i) the QC score, (ii) the widely adopted semiquantitative score determined from visual assessment (SV score) [17], and (iii) a computer assisted evaluation of the SV score (SC score). We also evaluated the agreement between the three scores and, on a small subset of quasi-simultaneous radiographs, their test–retest replicability.

### 2.2. Data

Data were taken from an earlier study that had recruited vertebral fracture patients (thus prone to AAC), for which lateral lumbar spine radiographs were available. Informed consent was acquired during routine follow-up visits for those patients who were still alive at the time of our study. The present study was approved by the local IRB (study ID: CE AVEC 695/2018/Oss/IOR) and registered (clinicaltrials.gov ID: NCT03839732).

### 2.3. Design

Given that our study had only a technical purpose (repeatability and reproducibility of AAC scores), the only inclusion criterion was the presence of AAC. Digital radiographs of 44 patients were selected from an initial list of 89 patients. Almost all patients were old (mean age 74 ± 8, age range 54–89, 38 patients older than 65), and most of them were females (35 vs. 9 males). The almost 50% prevalence of AAC was in good agreement with a previous, larger study on a similar cohort [25].

The operators were five experts from four medical specialties (one radiologist, one nephrologist, one internal medicine expert and two spine surgeons). Operators were deliberately chosen among different medical specialties to reflect the variety of clinicians confronting with AAC evaluation. Two operators (AB, MF) were already acknowledged experts in AAC evaluation. All operators initially were given the software and a guide to AAC evaluation. Indications on how to perform the AAC evaluation had been specified from the literature for the SV score, and by author MF (the inventor of the new score, and an expert of AAC evaluation) for the QC score. To train, all operators performed AAC evaluation on four radiographs (not included in the study), followed by a consensus meeting on how to best estimate SV score and use the software to measure QC score. The consensus meeting was guided by author AB, a radiologist and expert of AAC evaluation. No other interaction on the study took place between operators prior to the study end.

All operators performed the computer assisted evaluation of AAC twice, leaving at least one month time interval between assessment. Each radiograph was anonymized and randomized to two case series to avoid operator bias when re-evaluating each case. 

An additional test–retest evaluation was performed on eight (8) subjects for whom a second lumbar spine radiograph had been taken within two weeks from the first one. These second radiographs had been taken for clinical purposes, to verify upon symptoms referral by the patients whether other vertebral fractures had developed.

### 2.4. Software and Measurement Protocol

A specific software tool (named *Calcify2D*) was developed to assist the identification of AAC and the evaluation of the three scores: QC, SC, and SV. The software, written in C++, is based on the open-source framework ALBA (Agile Library for Biomedical Applications, https://github.com/IOR-BIC/ALBA). It features a graphical user interface through which the operator can perform measurements onto calibrated lateral spine radiographs, following a pre-defined workflow (Figure 1). The operator is guided through the following steps: (i) visual evaluation of AAC to determine the SV score; (ii) identification of vertebral bodies (from T12 to L5) through a simple guided user interaction; (iii) drawing of calcified tracts on the radiograph. The software then automatically reports the SV score, computes the QC according to the drawn calcified tracts, and back-calculates the SC score (see Section 2.5). To avoid any possibility of self-influencing, the software version used for the present study inhibited the report printout, so that the resulting scores were not available to the operators. Reports were printed and results analyzed only by non-operators (authors ES, GT, AA, FT) at the end of the data collection.

### 2.5. Output

SV score: discrete, measurement range 0–24 [17].

QC score (new): continuum, natively expressed in a 0–1 range. In this study QC was multiplied by 24 to achieve a 0–24 range and thus directly compare with the SV score output.

SC score: discrete, range 0–24. SC was automatically recalculated by the software from QC, according to SV score rules (subdivision in vertebral tracts, scoring of each anterior/posterior vertebral tract in tertiles). This was meant to uncouple, in the analysis of the results, the contribution of computer-assisted vs. visual evaluation from that of continuum vs. semi-quantitative scoring system.

### 2.6. Statistical Analysis

Intra-operator repeatability between the two data series of each score was quantified through Bland–Altman (B-A) plots and regression analysis. Data were analyzed per operator, and then pooling data among all operators.

Inter-operator repeatability was quantified for both repetitions of each score by computing the Intra-class Correlation Coefficient (ICC) using a two-way random effects model, considering single measurements. ICC permitted a straightforward comparison with existing repeatability studies on the SV score.

A comprehensive, sample-independent estimate of operator repeatability was determined by computing the Standard Error of the Measurement (SEM). This allowed us also to determine the MDD [24].

To preliminarily assess replicability (i.e., whether the scores are consistent when evaluated from two different, but quasi-simultaneous radiographs) a paired non-parametric Wilcoxon test was performed on the test–retest data, using the second repetition to minimize any possible learning effect.

Finally, the agreement of the scores was estimated through Bland–Altman plots of the differences between the scores.

All analyses were conducted with a commercially available statistical software (SPSS for Windows, Version 28, IBM, Chicago, IL, USA).

## 3. Results

Two radiographs were judged as non-compliant with the minimum image quality standards. This happened independently in both repetitions by the two most experienced operators. We therefore present results for 42 subjects. Data are reported for 204 paired measurements: from the complete set of 210 measurements (i.e., five operators on 42 subjects) two were excluded because of software error and four because of incomplete operator measurement.

### 3.1. Intra-Operator Repeatability

According to B-A plots (Figure 2, pooled data from all operators), all scores had equal and almost null average difference between repetitions. Also, no trends of differences with increasing mean values of the scores were visible. But notably, score variations were smaller in QC (95% Confidence Interval (CI) 6.1, i.e., one quarter of the whole 0–24 range) than in SV (CI 11.6, close to half the whole range) (Figure 2). The CI for SC score was 9.2, in between QC and SV. Results were consistent among the operators (B-A plots per operator in the Appendix A).

Similarly, regression analyses returned a higher determination coefficient (R^2^, i.e., how much variance is common to the two repetitions) for QC score (R^2^ = 0.79 when pooling data from all operators) compared to SV score (R^2^ = 0.67), with SC score in between (R^2^ = 0.70) (Figure 2). Consistently with a higher repeatability of computer assisted techniques, when analyzed per operator (plots available as Appendix A) the range of R^2^ among operators was narrower for QC (0.78–0.89) and SC (0.69–0.77), compared to SV (0.61–0.82).

### 3.2. Inter-Operator Repeatability

Data are reported in Table 1. In the first repetition, QC showed a higher Intra-class Correlation Coefficient (0.81 vs. 0.65) and a narrower 95% CI compared to SV. SC score had an intermediate ICC (0.75) with a CI range closer to QC. In the second repetition, QC gained few percentage points in ICC (0.84) while SV almost closed the gap reaching an ICC of 0.82, and ICC for SC was 0.79. Confidence intervals in the second repetition were very similar for QC and SV scores.

### 3.3. Minimum Detectable Difference (MDD)

Considering all measurements pooled together, the SEM of QC was significantly lower than that of SV (1.15 vs. 2.09, i.e., 5% vs. 9% of the 0–24 range). This led to an MDD of 2.98 for QC, and 4.01 for SV. Operationally, as SV is a discrete score, the MDD has to be rounded to the closest integer, i.e., 4. The SEM for SC, the semi-quantitative score back-calculated from the calcifications traced in the software, was 1.71, closer to SV than to QC. In fact, the MDD for SC when rounded to the closest integer was 4, the same as SV.

### 3.4. Test–Retest Reproducibility

We report results for: (i) seven (out of eight) cases, as one radiograph in the test–retest group was among those excluded from the repeatability analysis; (ii) the second repetition, where inter-operator repeatability for QC and SV was similar, thus by-passing the learning artefact observed for SV score between first and second repetition. On the small dataset available, the evaluation of AAC on the second series of radiographs was undistinguishable from that on the first series according to a paired non-parametric Wilcoxon test (*p*-value > 0.10 for all scores). *p*-value was the highest for QC (*p* = 0.95), followed by SC (*p* = 0.45) and SV (*p* = 0.11).

### 3.5. Agreement between Scores

The estimate of AAC obtained through the quantitative computer-assisted QC score were apparently different from those obtained with the semi-quantitative visual SV score, while this latter appeared similar to the semi-quantitative computer-assisted score (SC) (Table 2). The B-A plot of the SV-QC differences (here reported for the second repetition to bypass any learning artefact) highlighted: (i) an asymmetry with respect to zero, as the mean value of the differences was close to four in the 0–24 range, indicating that generally the SV score was higher than the QC score; (ii) an increasing trend of differences with the measured value; (iii) a wide 95% confidence interval (CI = 9.6 in the 0–24 overall range).

The comparison of QC and SV scores with the SC score helped understand different contributions to the observed differences (Figure 3). The SC-QC differences (isolating the contribution of quantitative continuum vs. semiquantitative discrete score) looked similar to SV-QC differences. In fact, SC-QC also showed a clear increasing trend of differences when the mean score value increased, and an average difference over three. The 95% CI was narrower (CI = 4.9 in the 0–24 range, almost halved with respect to the 9.6 CI for SV-QC), but this was expected because both scores originated from the same computer-assisted measurements. The SV-SC differences (isolating the contribution of computer-assisted vs. visual estimation) showed an almost null mean/median value, despite a significant dispersion (CI = 9.2) and a slight increasing trend of differences with mean score value.

## 4. Discussion

The original proposal of our study is a new quantitative vascular calcification score from lumbar spine radiographs, based on the relative length of AAC with respect to overall AA length. The score evaluation is assisted by a simple software application that permits interactive visualization and tracing of the calcified tracts visible in the radiographs. The results corroborate the hypothesis that this new score (QC score in this study) can improve the precision of the currently most used radiographic score for the estimate of AAC. 

The QC score achieved a better intra-operator repeatability than the usual semiquantitative score (SV score in this study): between repetitions, the 95% CI of differences was almost halved in QC, and R^2^ was 0.79 for QC vs. 0.67 for SV.

Inter-operator repeatability as measured by ICC was instead equivalent for the two scores, but this happened only in the second repetition of measurements. The lower values of ICC seen for all scores in the first repetition were likely due to learning artefacts, attributable both to the use of a new software (for QC and SC score) and to the different level of experience of some operators in evaluating AAC (for all scores). However, the QC score achieved an ICC close to 0.8 already in the first repetition, and the improvement in ICC between repetitions was significantly smaller for the computer assisted scores. There is thus evidence that a computer assisted procedure may be useful when AAC evaluation has to be performed by non-expert clinicians. 

The new QC score achieved also a lower MDD, reduced by one fourth (approximately from 4 to 3 in the 0–24 scale) with respect to SV. This is a crucial aspect in the evaluation of a score or measurement as MDD discriminates noise from actual differences [24]. Upcoming longitudinal clinical studies relating AAC to clinical endpoints will lead to the definition of clinically relevant differences, which will likely be pathology-specific. MDD should be a crucial figure when designing and interpreting longitudinal studies, especially in light of the shortening of follow-up times seen in the CKD field [22,23]. However, to our knowledge it has never been considered prior to our study.

The main contribution to the improved precision performance of QC score is the quantitative, continuum nature of the score. Evidence of this statement comes from the results of the SC score, as the SEM and MDD of SC score are closer to that of SV than to that of QC. The SC score was exactly meant to decouple the contribution of computed assisted vs. visual procedure and continuum vs. discrete scoring system when interpreting the differences between QC and SV. Overall, the higher precision of the new QC score prelude to the possibility of a considerable reduction in the patients number required to address research and/or clinical hypotheses. For example, a smaller sample size would be needed to detect a longitudinal change in VC progression and/or to assess the effectiveness of a specific intervention. This suggests a potential for important savings in research costs. 

Ours appears the most comprehensive study to date on the repeatability of AAC scores from 2D radiographic projections (lateral lumbar spine radiographs or DXA). Several other studies reported repeatability metrics for the SV score (see Table 3 for a brief review). Still, no other indicators are usually available than ICC, or concordance statistics on patient classification according to a threshold [17,26,27,28,29,30,31,32]. Our inter-operator ICC results for the SV score are generally lower than other studies: in the first repetition the ICC we report (0.65) is similar only to that of Toussaint et al. [28], which, however, stands out from all other reports. Moreover, considering the second repetition, our ICC (0.84), although good, is lower than the 0.89–0.96 range found in other studies. This difference may be explained by the fact that no other study featured more than two operators, while we included five operators from different specialties, to better reflect the operational environment in which AAC evaluation may happen. We believe our experimental design is robust to evaluate the repeatability and reproducibility of AAC scores. In fact, it may represent a worst-case scenario where operators from different medical specialties and with different levels of experience from two different institutions perform, with no interactions between them, two repetitions on randomized data series.

Another notable and original, although preliminary finding, is the robustness of QC to radiograph differences. Our small test–retest experiment reported small and largely non-significant QC score differences when using short-term radiographs repetitions, although image quality could differ a lot between the two repetitions. Differences were negligible also for the SV score, although the *p*-value was smaller (0.11), so that a larger sample would probably be needed to confirm the result. Actually, the very small sample size is the main limitation of our test–retest experiment.

Despite QC and SV score are both intended to evaluate AAC extent, the agreement analysis (Figure 3) speaks for little concordance of the two scores: (i) the average difference (evaluated on the second repetition, and therefore likely free from any learning bias) is well over the SEM of both scores, and (ii) an increasing trend in the difference with calcification severity is clear. The QC-SC Bland–Altman plot (quantitative vs. semiquantitative, but both computer-assisted) suggests the discrete nature of the score (according to the SV score rules) matters more than the computer-assisted vs. visual assessment in determining differences of the AAC estimate. This hypothesis was indirectly confirmed by the SV-SC plot, where both semiquantitative plots had a mean difference very close to zero. A simplified working example may explain how the QC-SV difference is generated. Imagine a relatively short calcified tract, extending over two intervertebral spaces (or two consecutive short tracts in two different spaces): it would count 2/24 in the discrete SV score, but possibly far less than 2/24 in the continuum QC score. If this condition were repeated over different intervertebral spaces (and anterior–posterior walls) the two scores would diverge. An inherent difference in SV and QC score thus emerges from these results, but we cannot presently tell how this reflects into the judgement of score severity. In fact, any concordance test in patient classification would be hard to perform, as (i) there is no consensus on SV score severity thresholds (they range from 4 [33] or 5 [34], to 8 [20], to even 12 [35] in the 0–24 scale); (ii) severity thresholds for QC score threshold are still unknown and cannot be taken from SV as the two score seem to measure differently. There is, therefore, the need to test the new QC score in clinical cohorts and define absolute or progression severity thresholds by relating it to clinical outcomes. To this aim, a prospective study in chronic kidney disease is in preparation. In that study, we may also attempt the extension of the principle of our new score to other narrower but important vessels, such as the iliac arteries.

The main limitation of the present study is its retrospective nature. We made opportunistic use of radiographs taken for fracture detection or planning of vertebroplasty treatment. However, repeatability was satisfactory even using sub-optimal images. The test–retest data corroborate this hypothesis, although on a very small sample. Another limitation is the absence of validation of QC score against a gold standard (CT). Three studies from two groups have tried to relate SV from DXA to AAC measurements from CT [28,36,37]. They reported a relatively good sensitivity/specificity of DXA in the detection of AAC (improved when limiting to severe AAC cases) but an only moderate correlation with CT-based estimates (range of correlation coefficients r from 0.51 to 0.60, i.e., variance explained up to as low as 36%) [36,37]. A quantitative score such as QC may have a bigger potential in correlating with CT-based AAC estimates, besides the unavoidable limitations of 2D vs. 3D imaging. Site-specific comparison with a CT-based score equivalent to CAC may be the method of choice to validate QC score estimates, and is within our plans for further studies [38]. Finally, the software user-experience in the computer-assisted evaluation of AAC could be improved, as now the workflow is rather slow (requiring 10 min on average to complete one evaluation). This is an important issue in clinical settings where time availability is generally scarce. Automatic detection of AAC would solve this issue and also eliminate any operator bias. Preliminary works in this direction exist but were focused on discriminating low vs. high-risk patients using the semiquantitative discrete score as a gold standard reference [31,35]. In our view, there would be an even stronger indication for the application of machine learning/artificial intelligence approaches to automatically determine the quantitative and continuum QC score.

## 5. Conclusions

In summary, we developed a new quantitative continuum computer-assisted measurement for scoring AAC. We demonstrated it has better intra-operator repeatability, lower influence of operator experience, and most of all lower minimum detectable difference (MDD) when compared to the most-used semiquantitative scoring system.

These results candidate this new score to reliably evaluate progression of AAC using 2D radiographic projections in shorter terms than currently possible. However, as the new quantitative score appears to behave differently than the widely used semiquantitative score, larger studies on clinically relevant cohorts are needed to define its actual association with clinical outcomes, including definition of possible severity thresholds.

## Figures and Tables

**Figure 1 nutrients-14-04276-f001:**
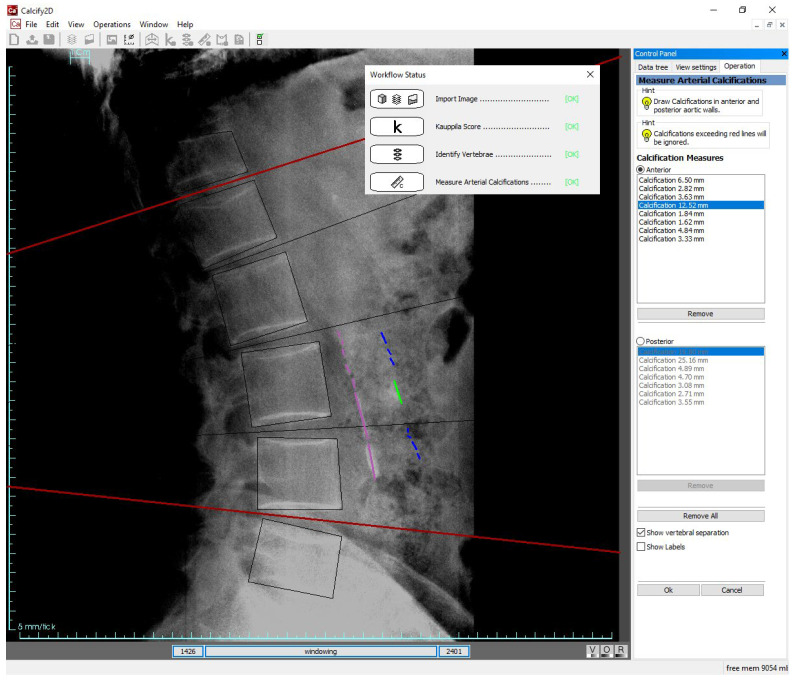
Scoring AAC with Calcify2D software: after having imported the radiograph, estimated the SV score, and identified the L1–L5 vertebral bodies (steps not shown on the image, but traced in the workflow status box present on the screen), the operator draws the calcified tracts on the anterior and posterior walls of the abdominal aorta by dragging the mouse on the screen. Grayscale windowing and image zooming are permitted. Intervertebral separations and length of calcified tracts can be shown or hidden. Calcified tracts exceeding red lines (representing T12-L1 and L4–L5 boundaries) are ignored in the calculation of the scores.

**Figure 2 nutrients-14-04276-f002:**
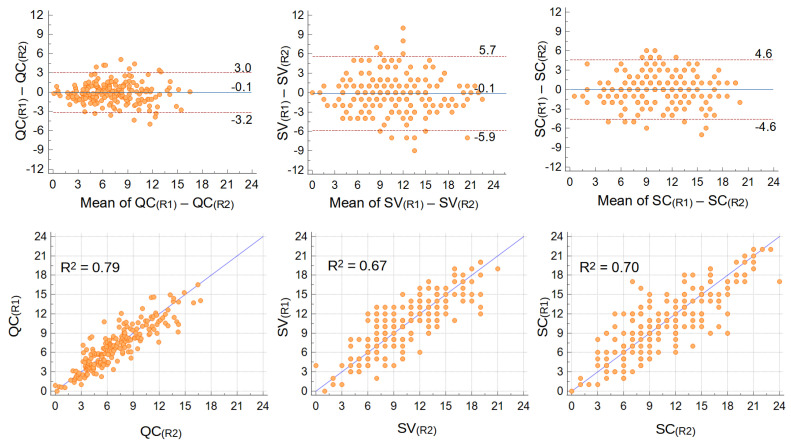
Intra-operator repeatability: Bland–Altman (upper) and regression (lower) plots of differences between repetitions of QC (left), SV (center), and SC (right) scores. Mean differences and 95% CI are shown in B−A plots, trend-line and R^2^ in regression plots. Data from all operators were pooled (*n* = 204 repeated measurements).

**Figure 3 nutrients-14-04276-f003:**
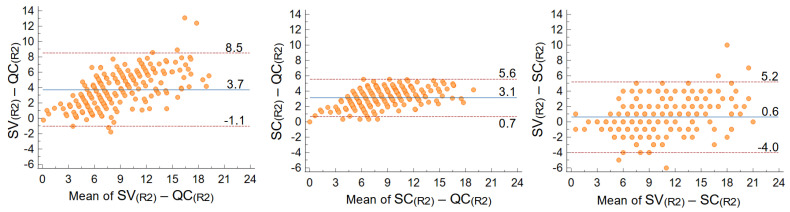
Agreement between scores: Bland–Altman plots of SV−QC (left), SC−QC (center), and SV−SC scores in the second repetition of measurements (R2). Mean differences and 95% CI are shown on the plots.

**Table 1 nutrients-14-04276-t001:** Inter-operator repeatability: intra-class correlation coefficients (ICC) and their 95% CI.

Repetition	Score	ICC	ICC 95% CI
R1	QC	0.81	0.65–0.90
	SV	0.65	0.43–0.80
	SC	0.75	0.57–0.86
R2	QC	0.84	0.74–0.90
	SV	0.82	0.74–0.89
	SC	0.79	0.68–0.87

QC: Quantitative Computer-assisted score (the new score presented in this study); SV: Semi-quantitative Visual score (the current standard); SC: Semi-quantitative computer-assisted score (recalculated from QC according to SV rules).

**Table 2 nutrients-14-04276-t002:** Median and interquartile range of the three AAC scores measured in the whole cohort.

Repetition	SV	SC	QC
R1	10 (7, 14)	10 (8, 13)	6.7 (4.3, 9.1)
R2	10 (7, 14)	10 (7, 13)	6.5 (4.3, 9.1)

QC: Quantitative Computer-assisted score (the new score presented in this study); SV: Semi-quantitative Visual score (the current standard); SC: Semi-quantitative Computer-assisted score (recalculated from QC according to SV rules).

**Table 3 nutrients-14-04276-t003:** Brief review of repeatability studies on radiographic AAC scores.

Publication	Source Image	Design	Size	Results
Kauppila et al., 1997 [17]	Radiographs	2 operators once	50	ICC inter = 0.93 and 0.96 (f.-up)
		1 operator twice	50	ICC intra = 0.98 and 0.96 (f.-up)
Schousboe et al., 2006 [26]	DXA and DR	2 operators once	57	ICC inter = 0.89 (DXA) and 0.92 (DR)
Bolland et al., 2010 * [27]	DXA	2 operators once	30	k inter = 0.87
		1 operator twice	100	k intra = 0.90
Toussaint et al., 2010 [28]	DXA	2 operators twice	44	ICC inter = 0.61 ICC intra = 0.94
Bazzocchi et al., 2012 * [29]	DXA and DR	2 operators twice	75	k inter = 0.71 (DXA) and 0.76 (DR) k intra = 0.95 (DXA) and 0.96 (DR)
Szulc et al., 2013 [30]	DXA	2 operators twice	76	ICC inter = 0.90 ICC intra = 0.95
Reid et al., 2021 [31]	DXA	2 operators once	77 **	ICC inter = 0.90
Lankinen et al., 2022 [32]	DR	2 operators twice	150	CV inter = 11.5% CV intra = 4.8%

* These studies reported repeatability in the classification above or below a given threshold and therefore used the kappa statistic rather than ICC. It is therefore difficult to compare their results with those of the current study. ** Sample size for Reid et al., 2021 was inferred from the text of the publication.

## Data Availability

The data that support the findings of this study are available on motivated request sent to the corresponding authors, [M.F. and E.S.]. The data are not publicly available due to restrictions because their containing information that could compromise the privacy of research participants.

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
