# Peer review of "A Novel Quantitative Computer-Assisted Score Can Improve Repeatability in the Estimate of Vascular Calcifications at the Abdominal Aorta"

_nutrients, 2022, doi:10.3390/nu14204276_

Round 1
Reviewer 1 Report
First, I congratulate the exciting topic and the idea of searching for a new score to reduce intra- and inter-observer variability in the computation of vascular calcifications at the abdominal aorta. As the population is increasingly ageing, so does the incidence of vascular calcifications. Thus, an earlier and more reliable marker/score could benefit large populations. The introduction is challenging to follow, and there is a loop from one idea to another without insisting on any of them. For somebody who is not an expert in the radiology field, it is difficult to understand precisely what is described (lines 65-77). Lines 80-81 – the same abbreviation is utilised with different meanings (ICC). It is mentioned that the follow-up was diminished from 25 to 3 to 1 year. What do the guidelines specify? And what are the ups and downs depending on the length of the follow-up? (line 84). Lines 99-104 belong to the template of the journal. You begin your study by mentioning that patients with vertebral fractures are prone to AAC (lines 117-118). The main limitation of the review that approached this link concluded that the main weakness of the research is that causality cannot be established due to the observational nature of the included studies. It is not mentioned if the study is prospective or retrospective and the way patients were recruited. It is mentioned that the only inclusion criterion (line 122) was the presence of AAC. There were no exclusion criteria? Although initially there were 89 patients, the final lot contained 44. And did the patients sign an informed consent for this study? The English language needs professional revising. To conclude, the manuscript is not very clear. It could be relevant to the field if revised and clarified. The experimental design is lacunar and exposes some problems. First, the five experts were not experts but doctors who encountered the AAC diagnosis. Second, their training was held by whom? Was it a radiologist? Was it the same person for all five? It is a valid question because, in their training, they could receive information in such a matter that it was very likely for all of them to interpret the images similarly. Third, for test-retest on the 8 cases, was it ethical? The second radiograph in less than two weeks in the same place, on the same patient, was performed. However, currently, the paper does not fit the aim and scope of the journal. “Nutrients will consider manuscripts for publication that provide novel insights into the impacts of nutrition on human health or novel methods for assessing nutritional status.” Adding a separate section to approach the impact of nutrition on AAC could make it more suitable. If you consider and find it appropriate, you could add these references to the list. https://www.ncbi.nlm.nih.gov/pmc/articles/PMC6500258/, https://www.mdpi.com/2075-4426/12/7/1120, https://www.mdpi.com/2077-0383/11/5/1157, https://nutritionj.biomedcentral.com/articles/10.1186/s12937-022-00782-0, https://www.mdpi.com/1660-4601/18/10/5444.
Reviewer 2 Report
Fusaro et al. presented a manuscript describing a novel computer-assisted score for quantification of vascular calcification in the abdominal aorta.
Major Comments:
1) Cross reference to supplement. material (lines 206 and 216): suppl. material was not available.
2) Please include more information for patient characteristic: CKD, metabolic syndrome, how many old patients.
3) Please include a figure/table to show the intensity of vessel calcification throughout your study cohort.
4) Is your score presented for the abdominal aorta also transferable to other vessels? Please discuss.
Minor Comments:
1) 4 Authors have the sign * for corrensponding author. Do you mean contributed equally for the first two and last two authors?
2) Lines 99-104 should be removed.
3) Please include a footnote for Table 1 giving description of abbrevariations used: OC, SV, SC.
4) Consider to rephrase very long sentences in the manuscript.
5) Please check abbreviations: use consistently upon first use.
6) Please check the reference list: e.g., ref. 1 is incomplete (journal is missing, author number)
Round 2
Reviewer 2 Report
Thanks for the revised version.